# Betulinic Acid-Mediated Tuning of PERK/CHOP Signaling by Sp1 Inhibition as a Novel Therapeutic Strategy for Glioblastoma

**DOI:** 10.3390/cancers12040981

**Published:** 2020-04-15

**Authors:** Wei-Lun Lo, Tsung-I Hsu, Wen-Bin Yang, Tzu-Jen Kao, Ming-Hsiao Wu, Yung-Ning Huang, Shiu-Hwa Yeh, Jian-Ying Chuang

**Affiliations:** 1The Ph.D. Program for Neural Regenerative Medicine, College of Medical Science and Technology, Taipei Medical University and National Health Research Institutes, Taipei 11031, Taiwan; 12317@s.tmu.edu.tw (W.-L.L.); dabiemhsu@tmu.edu.tw (T.-I.H.); z18981026@tmu.edu.tw (W.-B.Y.); geokao@tmu.edu.tw (T.-J.K.); m120104031@tmu.edu.tw (Y.-N.H.); 2Division of Neurosurgery, Taipei Medical University-Shuang-Ho Hospital, New Taipei 23561, Taiwan; wum@tmu.edu.tw; 3TMU Research Center of Neuroscience, Taipei Medical University, Taipei 11031, Taiwan; 4TMU Research Center of Cancer Translational Medicine, Taipei Medical University, Taipei 11031, Taiwan; 5Cell Physiology and Molecular Image Research Center, Wan Fang Hospital, Taipei Medical University, Taipei 11031, Taiwan; 6Institute of Biotechnology and Pharmaceutical Research, National Health Research Institutes, Miaoli 35053, Taiwan; bau9763@nhri.edu.tw

**Keywords:** protein kinase RNA-like endoplasmic reticulum kinase, C/EBP homologous protein, betulinic acid, specificity protein 1, glioblastoma

## Abstract

Patients with glioblastoma are at high risk of local recurrences after initial treatment with standard therapy, and recurrent tumor cells appear to be resistant to first-line drug temozolomide. Thus, finding an effective second-line agent for treating primary and recurrent glioblastomas is critical. Betulinic acid (BA), a natural product of plant origin, can cross the blood–brain barrier. Here, we investigated the antitumor effects of BA on typical glioblastoma cell lines and primary glioblastoma cells from patients, as well as corresponding temozolomide-resistant cells. Our findings verified that BA significantly reduced growth in all examined cells. Furthermore, gene-expression array analysis showed that the unfolded-protein response was significantly affected by BA. Moreover, BA treatment increased activation of the protein kinase RNA-like endoplasmic reticulum kinase (PERK)/C/EBP homologous protein (CHOP) apoptotic pathway, and reduced specificity protein 1 (Sp1) expression. However, Sp1 overexpression reversed the observed cell-growth inhibition and PERK/CHOP signaling activation induced by BA. Because temozolomide-resistant cells exhibited significantly increased Sp1 expression, we concluded that Sp1-mediated PERK/CHOP signaling inhibition protects glioblastoma against cancer therapies; hence, BA treatment targeting this pathway can be considered as an effective therapeutic strategy to overcome such chemoresistance and tumor relapse.

## 1. Introduction

Glioblastoma (GBM) is the most common malignant primary brain tumor in adults, and it is characterized by an aggressive clinical course. The primary treatments for GBM are maximal surgical cytoreduction, concurrent chemoradiotherapy, and antivascular endothelial-growth-factor-targeted therapy [1,2]. Temozolomide (TMZ) is a first-line chemotherapeutic agent. Its small size and lipophilic properties facilitate its ability to efficiently cross the blood–brain barrier (BBB), allowing TMZ to be orally administered [3]. However, the prognosis of patients with GBM is still poor, and the five-year survival rate is less than 10%, even after radiation with TMZ [4]. The development of TMZ resistance is a major challenge; approximately 90% of patients suffer from disease recurrence within two years of treatment, regardless of their prior response to initial treatment [4]. Moreover, there is no standard therapeutic approach for management of tumor recurrence.

The endoplasmic reticulum (ER) is an important organelle responsible for the synthesis, folding, and assembly of soluble and membrane-bound proteins in a cell. However, environmental or intracellular factors may disrupt ER functions, causing the accumulation of unfolded proteins and induction of ER stress. Several functional factors are known to be involved in ER stress, including membrane-bound stress sensors, transcription factors, and heat-shock chaperone proteins (HSPs) [5]. Previous studies revealed that the ER chaperone glucose-regulated protein 78 (Grp78, also known as BiP) serves as a master regulator of ER stress and plays important roles in activating three major signaling events, initiated by inositol-requiring enzyme-1 (IRE1α), activating transcription factor 6 (ATF6α), and protein kinase RNA-like endoplasmic reticulum kinase (PERK), which may mitigate stress and restore ER homeostasis to promote cell survival and adaptation [6,7]. However, during prolonged ER stress, PERK signaling can also trigger proapoptotic signals via activation of the downstream C/EBP-homologous protein (CHOP) to promote apoptosis [8]. In human brain tumors, an increase in Grp78 levels is detected in malignant GBM compared with normal brain tissue [9,10]. Additionally, its overexpression mediates ER stress tolerance, and protects GBM cells against the cytotoxicity of TMZ [9,11]. Therefore, ER stress-related factors may have roles contributing to tumorigenesis and to the development of TMZ resistance in GBM. Additionally, altering the activities of these factors may be a potential strategy to overcome brain-tumor recurrence after initial therapy.

Betulinic acid (BA), a pentacyclic triterpenoid found in white-birch bark, has multiple biological effects against micro-organisms, viruses, parasites, and neoplasms. Importantly, BA exhibits selective cytotoxicity in cancer-cell lines, but not in normal cell lines [12]. For example, our previous study showed that BA affected the SUMOylation of the oncogenic specificity protein 1 (Sp1) transcription factor, thereby inhibiting lung-cancer growth [13]. Other studies further revealed that BA induces apoptotic cell death in human cancers, including head and neck, colon, breast, prostate, ovarian, and cervical cancers, by triggering the mitochondrial apoptotic pathway, but showed that normal cells and tissue are relatively resistant to BA [14]. Recently, our group further demonstrated that SYK023, a synthetic derivative of BA, attenuates lung-cancer growth and malignancy by increasing ER stress [15]. However, the mechanisms underlying the specific targets of BA during ER stress remain elusive. Furthermore, BA has been shown to cross the BBB; at 3 h after BA injection, its concentrations in brain and in plasma reached peak values of 38.6 ± 3.2 ng/g and 207.9 ± 34.5 ng/mL, respectively [16].

Therefore, in this study, we evaluated the applicability of BA as a potential therapeutic agent for the treatment of GBM, particularly in TMZ-resistant and recurrent GBM, through inhibition of ER stress tolerance.

## 2. Results

### 2.1. BA Selectively Targets Brain-Tumor Cells

We examined the pharmacokinetics of BA in U87MG, A172, P3, and P5 cells at different concentrations (20 and 40 μM) for 2 days. Results revealed that BA acted in a concentration-dependent manner to suppress the rate of proliferation in all GBM cell lines and corresponding TMZ-resistant cells (Figure 1A). Furthermore, we treated normal astrocytes, U87MG cells, and corresponding TMZ-resistant cells with BA at a high concentration (100 μM) for 2 days, and found cell viability did not change by BA in rat primary astrocytes (Figure 1B); consistently, a significant decrease in cell viability was observed in both U87MG parental and TMZ-resistant GBM cells.

### 2.2. BA Sensitizes Resistant GBM Cells to TMZ

We next examined whether BA sensitizes GBM cells to antineoplastic agent TMZ. In patient-derived TMZ-sensitive P3 cells, a lower concentration of BA (20 μM) inhibited tumor growth by approximately 25%, but did not result in additive anticancer effects when used in combination with TMZ treatment (Figure 2A). However, in TMZ-resistant GBM cells, BA at the same concentration was able to sensitize the resistant cells to a TMZ rechallenge (Figure 2B). Interestingly, in both TMZ-sensitive and -resistant GBM cells, 40 μM BA showed better tumoricidal activity than that of TMZ alone at concentrations of 100 μM or less (Figure 2A,B). Because cell death can be classified according to morphological features, we further investigated the morphology and size of resistant cells by light microscopy. The classic morphologies of apoptosis, including cell shrinkage and debris, were observed after combined treatment with BA and TMZ, but not after treatment with TMZ alone (Figure 2C), indicating that BA indeed enhanced the cytotoxicity and apoptosis of TMZ in malignant GBM cells.

### 2.3. BA Suppresses GBM Cell Growth via Inhibition of Sp1 Expression

Our previous studies showed that Sp1 expression is upregulated in high-grade brain tumors, and is significantly higher in TMZ-resistant cells than in parental GBM cells; however, inhibition of Sp1 protein expression restores the inhibitory effects of TMZ in malignant GBM cells [17,18,19]. Thus, we next determined whether BA treatment affected Sp1 expression in parental control (Figure 3A) and TMZ-resistant (Figure 3B) GBM cells. Results of Western blotting showed that Sp1 protein levels were downregulated in a concentration-dependent manner by BA in all cell lines. Subsequently, we found that Sp1 overexpression provided protection of GBM cells against BA treatment (Figure 3C).

### 2.4. BA Treatment Alters Expression of ER Stress-Related Genes

Sp1 is a transcription factor that plays a central role in regulating the expression of genes associated with pro-oncogenic activity [20]. Thus, attenuation of Sp1 expression by BA may alter the expression of various genes that regulate the malignant behaviors of GBM cells. To explore the mechanisms of tumor inhibition by BA and uncover novel therapeutic targets for GBM, we performed microarray analyses of RNA extracted directly from TMZ-resistant U87MG cells treated with dimethyl sulfoxide (DMSO) or 20 μM BA for 1 day, and the data were analyzed by Ingenuity Pathway Analysis (IPA) software. The top five canonical pathways are shown in Figure 4A. The unfolded-protein response (UPR), a signaling network that functions to alleviate ER stress, was most significantly affected by BA. Using cut-offs of fold changes greater than or equal to 2, and a *p* value less than or equal to 0.05, we found that 1341 genes were differentially expressed between BA- and non-BA-treated cells (Figure 4B). Among these genes, 21 ER-stress related genes were identified (Figure 4B), and the roles of these 21 genes are shown in Figure 4C. Subsequently, we examined the protein expression of ER stress-related genes. Western blotting showed that BA treatment in TMZ-resistant U87MG cells indeed increased protein levels of IRE1α, CHOP, activating transcription factor 3 (ATF3), protein phosphatase 1 regulatory subunit 15A (PPP1R15A)/growth arrest and DNA-damage-inducible 34 (GADD34), and homocysteine-responsive endoplasmic reticulum-resident ubiquitin-like domain member 1 protein (HERPUD1), and reduced the expression of X-box binding protein 1^unspliced^ (XBP1u), but did not alter the expression of DnaJ heat-shock-protein family member B1 (DNAJB1/HSP40), HSPA1A (HSP70-1A), HSPA6 (HSP70B), HSPA1L (HSP70-1L), HSP90B1 (Grp94), BiP (Grp78), and Sig-1R (Figure 4D).

### 2.5. Sp1 Plays Roles in Regulating UPR Activation

The PERK, IRE1α, and ATF6α signaling pathways relevant to UPR act as ER stress sensors by binding to ER chaperone Grp78, and remain inactive under normal conditions [21]. However, when unfolded proteins are accumulated in the ER, Grp78 preferentially binds to the unfolded proteins, leading to the release of PERK, IRE1α, and ATF6α. Subsequently, PERK and IRE1α undergo dimerization and autophosphorylation to initiate their downstream signaling pathways. ATF6α without Grp78 binding results in its translocation to the Golgi, where ATF6α is cleaved by proteases to become an active transcription factor. Here, we found that the phosphorylation and protein content of IRE1α, the cleaved/active form of ATF6α, and activation of the PERK/eIF2α/CHOP axis were increased after 2 days, but not 8 h, of treatment with BA, regardless of the presence or absence of TMZ (Figure 5).

Since BA treatment resulted in Sp1 inhibition (Figure 3) and UPR activation (Figure 4), we next examined the association between Sp1 expression and UPR activation using gene knockout and overexpression in GBM cells. As shown in Figure 5, we found a marked decrease in the cleaved/active form of ATF6α in Sp1-knockout U87MG cells. Furthermore, compared with wild-type cells, Sp1 silence showed a reduction in the BA-induced phosphorylation of IRE1α and a significant elevation in the BA-induced activation of the PERK/eIF2α/CHOP axis (Figure 5 and Appendix A). However, Sp1 overexpression increased basal levels of phospho-IRE1α and cleaved ATF6α and caused a significant reduction in the BA-induced expression of phospho-PREK, phospho-eIF2α, and CHOP proteins, suggesting that Sp1 played critical roles in promoting both the activation of the IRE1α/ATF6α pathways and the inhibition of the PERK/eIF2α/CHOP axis in GBM cells.

### 2.6. BA Triggers Apoptosis and DNA Damage in TMZ-Sensitive and -Resistant GBM Cells

CHOP is a key mediator of ER stress-induced apoptosis [8]. Thus, we investigated induction of apoptosis using propidium iodide (PI)/Annexin V double-staining in combination with flow cytometry. Results (Figure 6A) revealed that BA induced apoptosis in both parental and TMZ-resistant GBM cells in a concentration-dependent manner. We also examined apoptosis-related biomarkers, and verified that BA increased poly-A ribose polymerase (PARP) cleavage, caspase-3 activation, and p53 phosphorylation in U87MG cells (Figure 6B). Because Sp1 protects GBM cells from DNA damage induced by oxidative stress [17], and BA attenuated Sp1 expression (Figure 3), we next examined DNA damage marker γH2AX and found that BA treatment resulted in concentration- and time-dependent increases in γH2AX expression in U87MG, A172, P3, and P11 GBM cells (Figure 6C–E). Therefore, these results suggested that BA induced cell death via apoptosis and DNA damage.

### 2.7. BA Inhibits GBM Growth and Improves Survival In Vivo

To evaluate the in vivo effects of BA, a TMZ-resistant xenograft model was generated by subcutaneous injection of TMZ-resistant U87MG cells. Mice were given intraperitoneal injections of BA (25 mg/kg) three times a week for 1 month, and the growth rate of xenografts was significantly slower in BA-treated mice than in the control group (Figure 7A). In parallel, 1 month after treatment with BA, we detected a decrease in tumor weight (~50%; Figure 7B). Furthermore, we established an orthotopic GBM animal model, and tumor volumes in mouse brains were quantified by bioluminescent imaging using an in vivo image system (IVIS). Results (Figure 7C–E) revealed that BA reduced tumor size and prolonged survival in mice in a dose-dependent manner.

## 3. Discussion

In this study, we showed that BA selectively targeted malignant brain cells, including both chemosensitive and -resistant GBM cells. Furthermore, BA treatment attenuated Sp1 expression in resistant cells and induced ER-mediated UPR activation, including prosurvival signaling (IRE1α, XBP1, and ATF6α [22,23]) and prodeath signaling (PERK, CHOP, ATF3, and GADD34 [8,22,23]). However, Sp1 overexpression enabled the alteration of UPR signaling, particularly the attenuation of BA-induced activation of the PERK/CHOP axis, thereby reversing the inhibitory effects of BA on GBM cell growth. Interestingly, a dramatic increase in Sp1 expression was observed in both TMZ-resistant and cancer stemlike GBM cells in our previous studies [17,24], and BA could induce proteasome-dependent degradation of Sp1 by increasing its SUMOylation [13]. On the basis of these findings, we concluded that high levels of Sp1 in malignant/stemlike GBM cells modulated the balance between prosurvival and prodeath signals, thereby protecting these cells against stress conditions and cancer therapies (Figure 8). Thus, BA may have novel therapeutic applications in the treatment of GBM through Sp1 inhibition and tuning the activation of PERK/CHOP signaling (Figure 8).

Sp1 is a zinc finger transcription factor that binds to guanine–cytosine (GC)-rich (GGGGCGGGG) sequences in the promoter of target genes, and stabilizes the transcription-initiation complex on the promoter to activate gene expression [25]. In tumorigenesis, Sp1 facilitates cancer progression by activating the expression of many oncogenes that play pivotal roles in the metabolism, proliferation, and metastasis of various cancer cells [26]. Previous studies showed that increased levels of Sp1 are related to resistance to chemotherapy drugs, clinical recurrence, and tumor progression from low- to high-risk in prostate cancer [26,27]. In our previous studies, we further revealed that stress-activated transcription factor Sp1 upregulates stemness genes (i.e., *BMI1* and *NANOG*) and cellular protectors (i.e., *CYP17A1* and *SOD2*) in GBM cells, contributing to TMZ resistance [17,18,24]. Here, we also found that Sp1 linked UPR signaling and ER homeostasis to therapeutic resistance in GBM cells, whereas BA treatment switched UPR signaling from prosurvival to proapoptotic in GBM cells by decreasing Sp1 expression. Dauer et al. conducted a similar study and showed that Sp1 downregulation mediated by mithramycin leads to chronic ER stress by disrupting the homeostatic mechanism in pancreatic cancer [28]. However, mithramycin (C_52_H_76_O_24_, molecular weight: 1085.1 g/mol) is not a small molecule (molecular weight ~400 g/mol) and is unable to cross the BBB by free diffusion, limiting its applications in the treatment of brain tumors. In contrast, BA (C_30_H_48_O_3_, molecular weight: 456.7 g/mol) is small enough to pass through the tight junctions of the BBB [16] and is a selective Sp1 inhibitor [13,29]. Moreover, in the current study, we further confirmed that BA could effectively suppress GBM xenograft growth in an orthotopic mouse model and identified the PERK/CHOP signaling pathway as a potential target for Sp1.

TMZ is known to cause O(6)-methylguanine, a lethal DNA lesion, in mitochondrial and nuclear genomes, resulting in subsequent oxidative stress [30] and ER stress [31,32] in tumor cells. In our in vitro cell model, although TMZ significantly reduced cellular proliferation and survival immediately following treatment, cell viability commonly recovered after about 1 month of TMZ treatment. Similarly, in clinical outcomes among patients with GBM, approximately 90% of patients suffer from disease recurrence within 2 years of chemoradiotherapy, regardless of initial response [4]. Activation of such cellular stress responses, including DNA damage-response systems [33], antioxidant enzyme systems [17,34], and UPR signaling pathways [32], seems to promote survival and resistance in response to TMZ-induced cytotoxicity. Interestingly, previous studies indicated that Sp1 plays an important role in induction of DNA repair genes (i.e., *MGMT* and *RAD51* [35]) and antioxidant genes (i.e., *SOD2* [17]) in response to TMZ therapy. Furthermore, our current results also clarified that Sp1 modulated UPR activation. Thus, these results suggested that TMZ-induced cytotoxicity combined with BA-mediated inhibition of Sp1 and Sp1 downstream cellular stress responses could be a potential therapeutic strategy for GBM.

## 4. Materials and Methods

### 4.1. Cell Preparation

Human cell lines A172 and U87MG were obtained from the American Type Culture Collection (Manassas, VA, USA). Human primary GBM cell lines P3 and P11 were obtained according to a protocol approved by the Taipei Medical University (TMU) Internal Review Board (approval no. 201006011 and 201402018). Cells were incubated in Dulbecco’s modified Eagle’s medium (Thermo Fisher Scientific, Waltham, MA, USA) with 10% fetal bovine serum (Thermo Fisher Scientific), 100 U/mL penicillin, and 100 μg/mL streptomycin. TMZ-resistant cells were maintained in the same culture medium containing TMZ (Sigma-Aldrich, St. Louis, MO, USA). CRISPR was used to generate the Sp1-knockout U87MG cell line. Plasmid pEGFP-Sp1 encoding green fluorescent protein (GFP)-tagged full-length Sp1 was transfected into U87MG cells to generate the Sp1-overexpression U87MG strain. All cells were incubated in a humid atmosphere at 37 °C.

### 4.2. Examination of Cell Viability

Cells (2 × 10^5^ cells/well) were seeded in 24-well culture plates. After overnight incubation, cells were treated with different concentrations of drugs as indicated for 48 h. Next, 3-(4,5-dimethylthiazol-2-yl)-2,5-diphenyltetrazolium bromide (MTT) assays (Sigma Aldrich) were used to exam cell viability and cytotoxicity, as previously described [17].

### 4.3. Western Blotting

Protein samples were separated by sodium dodecyl sulfate polyacrylamide gel electrophoresis, and transferred to polyvinylidene difluoride membranes (Bio-Rad Laboratories, Hercules, CA, USA). Western blotting was performed as previously described [36]. The following primary antibodies were used: anti-Sp1 (Merck Millipore, Darmstadt, Germany), anti-glyceraldehyde 3-phosphate dehydrogenase (Proteintech, Rosemont, IL, USA), anti-ERN1/IRE1α (GeneTex, Irvine, CA, USA), anti-XBP1 (GeneTex), anti-DDIT3/CHOP (Cell Signaling Technology, Danvers, MA, USA), anti-ATF3 (GeneTex), anti-PPP1R15A/GADD34 (GeneTex), anti-HERPUD1 (GeneTex), antisigma receptor (Sig-1R; Santa Cruz Biotechnology, Dallas, TX, USA), anti-DNAJB1/HSP40 (GeneTex), anti-HSPA1A/HSP70-1A (GeneTex), anti-HSPA6/HSP70B (GeneTex), anti-HSPA1L/HSP70-1L (GeneTex), anti-HSP90B1/Grp94 (GeneTex), anti-BiP/Grp78 (BD Biosciences, San Jose, CA, USA), antitubulin (Proteintech), antiphospho-IRE1α (Ser724; Novusbio, Centennial, CO, USA), anti-ATF6α (Novusbio), anti-eIF2α (GeneTex), antiphospho-eIF2α (Ser51; Cell Signaling Technology), anti-PARP (Cell Signaling Technology), anticleaved caspase-3 (Cell Signaling Technology), antiphospho-p53 (Ser15; Cell Signaling Technology), anti-γH2A.X (Ser139; Abcam, Cambridge, MA, USA), antiphospho-ATR (Ser428; Cell Signaling Technology), and antiphospho-Chk1 (Ser317; Cell Signaling Technology). Original uncropped blots are shown in Appendix A.

### 4.4. Apoptosis Assay

Apoptosis was measured by staining cells with Annexin V-FITC (Abcam) and PI solution, followed by flow-cytometry analysis [37]. PI-negative/Annexin V-negative cells were considered healthy, PI-negative/Annexin V-positive cells were considered apoptotic, and PI-positive/Annexin V-positive cells were considered late-apoptotic.

### 4.5. Microarray Analysis

TMZ-resistant U87MG cells were treated with DMSO or 30 μM BA for 24 h, and total RNA was extracted using TRIzol reagent (Thermo Fisher Scientific). Subsequently, gene-expression analysis was performed using a SurePrint G3 Human Gene Expression 8x60K Microarray by Welgene Biotech (Taipei, Taiwan). Genes with at least 2-fold changes in expression and a signal-to-noise ratio threshold of greater than 1.0 between DMSO- and BA-treated groups were identified. Furthermore, functional analysis was performed using IPA (Qiagen Bioinformatics, Denmark).

### 4.6. In Vivo Animal Model for GBM

All experiments and animal care were conducted in accordance with the guidelines and under the supervision of the Institutional Animal Care and Use Committee (approval no. LAC-2017-0464), TMU (Taiwan). Male immunodeficient NOD.CB17-Prkdc^scid^/JNarl (NOD/SCID) mice (8 weeks old; BioLASCO, Taipei, Taiwan) were maintained at the animal facility of TMU. For subcutaneous inoculation, 2 × 10^6^ TMZ-resistant cells were injected into the right flanks of mice as described previously [17]. For the orthotopic GBM xenograft model, mice were anesthetized with isoflurane gas and fixed on a stereotactic device. A linear incision was made over the scalp, and a small craniotomy was made over the right lower quadrant of the skull. U87MG-Luc-GFP cells (2.5 × 10^5^), dual labeled with luciferase and GFP, were injected into the brain area at ~3 mm depth from the brain surface. On Day 4 after transplantation, mice were assigned to three groups: (1) mice in the subcutaneous (*n* = 6) and orthotopic (*n* = 6) xenograft models injected with vehicle (20% DMSO, 10% Tween 80, and 70% phosphate-buffered saline) intraperitoneally, defined as the control DMSO group; (2) mice in the orthotopic (*n* = 6) xenograft model injected with BA (12.5 mg/kg) intraperitoneally, defined as the low-dose BA group; and (3) mice in the subcutaneous (*n* = 6) and orthotopic (*n* = 6) xenograft models injected with BA (25 mg/kg) intraperitoneally, defined as the high-dose BA group. Tumor size in the orthotopic xenograft model was examined using an IVIS twice per week. After being anesthetized, mice were injected with luciferase substrate solution intraperitoneally and then transferred to the IVIS chamber for image acquisition.

### 4.7. Statistical Analysis

All experiments were performed more than three times, and data were analyzed using Statistical Product and Service Solutions (SPSS) software. Results of MTT assays were evaluated by repeated-measure analysis of variance (ANOVA). Statistical analysis of protein expression and animal tumors was performed using two-way ANOVA. Survival analysis was performed using Kaplan–Meier curves and analyzed by Tarone–Ware tests. Results with *p* values of less than 0.05 were considered significant.

## 5. Conclusions

On the basis of current and previous findings, high levels of Sp1 in malignant/stemlike GBM cells modulated the balance between prosurvival and prodeath signals, thereby protecting GBM cells from stress conditions and cancer therapies. However, BA may represent a novel therapeutic strategy for the treatment of GBM owing to its effect on Sp1 inhibition and tuning the activation of PERK/CHOP signaling.

## Figures and Tables

**Figure 1 cancers-12-00981-f001:**
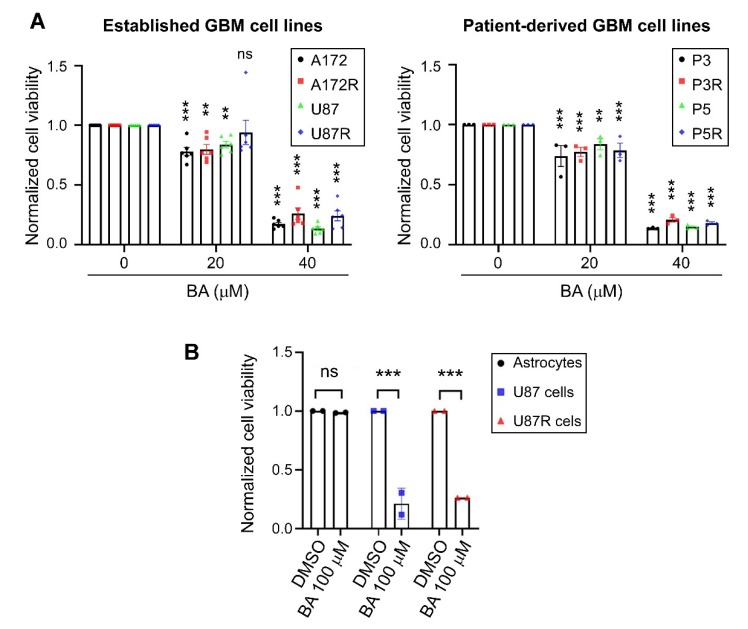
Betulinic acid (BA) suppresses tumor growth and survival in both parental and temozolomide (TMZ)-resistant (R) glioblastoma (GBM) cells, but not normal astrocytes. Cells were treated with 20, (**A**) 40, (**B**) or 100 μM of BA for 2 days. Dimethyl sulfoxide (DMSO) was used as vehicle control. After treatment, cell viability was determined by 3-(4,5-dimethylthiazol-2-yl)-2,5-diphenyltetrazolium bromide (MTT) assay. Data presented as means ± standard deviations (t-Test: ** *p* < 0.01 and *** *p* < 0.001; ns: not significant).

**Figure 2 cancers-12-00981-f002:**
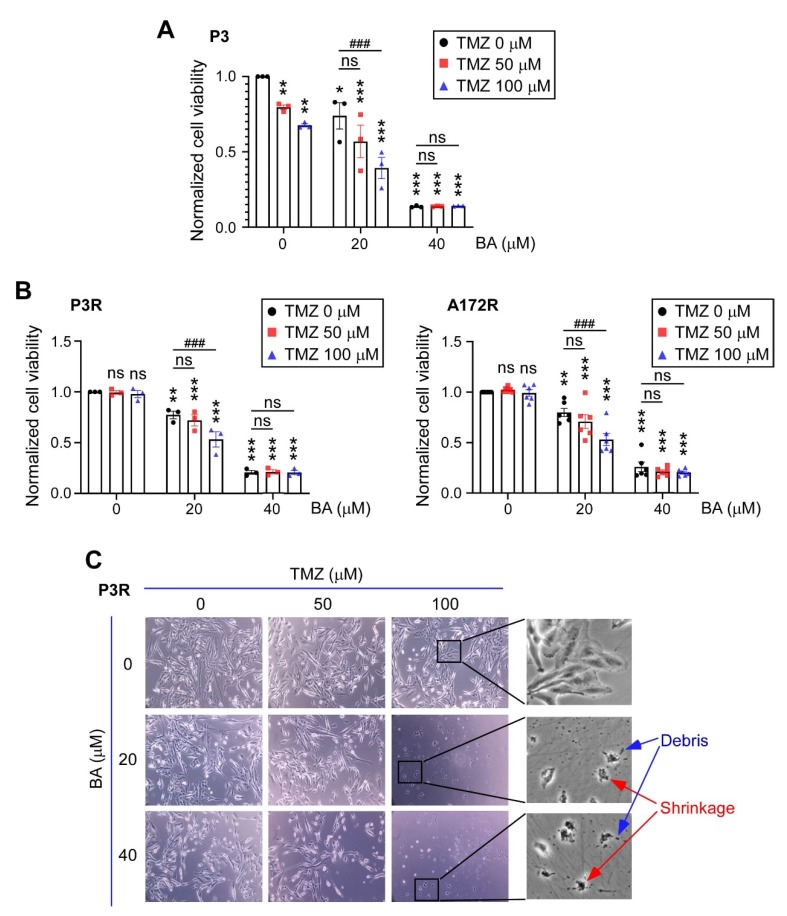
BA enhances TMZ antitumor effects in malignant GBM cells. (**A**) TMZ-sensitive P3, (**B**) TMZ-resistant P3R/A172R, and TMZ-resistant P5R GBM cells were treated with TMZ and/or BA at indicated concentrations for 2 days. (**A**,**B**) Cell viability of P3, P3R, and A172R cells determined by MTT assay. Data presented as means ± standard deviations (t-Test: * *p* < 0.05, ** *p* < 0.01, and *** *p* < 0.005 vs. nontreatment control; ^###^
*p* < 0.005 vs. TMZ-alone group; ns: not significant). (**C**) Representative images of P3R cells (original magnifications ×100 [left panels] and ×400 [right panels]).

**Figure 3 cancers-12-00981-f003:**
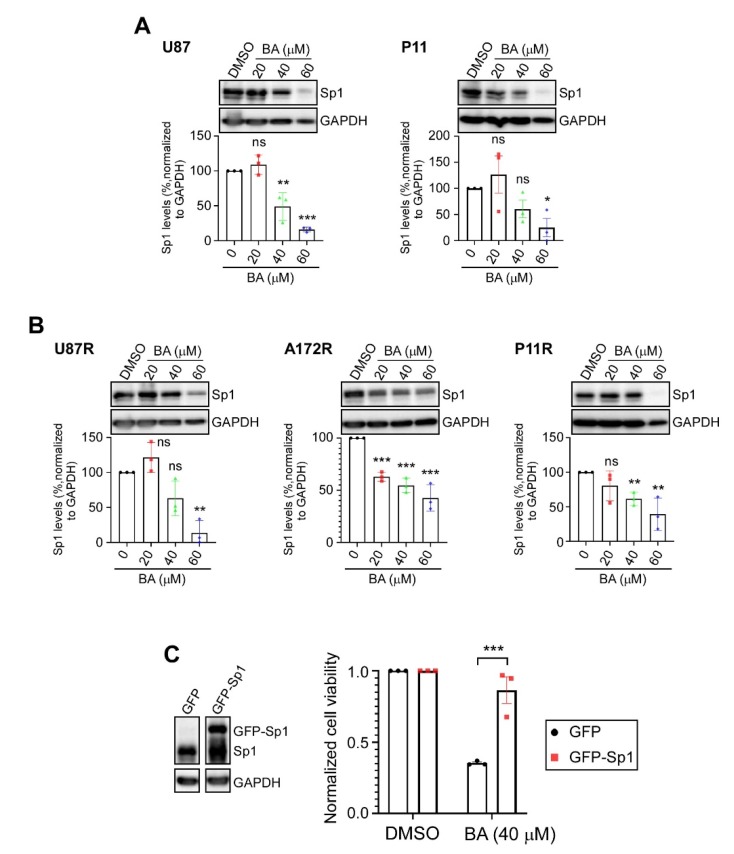
BA reduces Sp1 levels in GBM cells. (**A**,**B**) Cells treated with different concentrations of BA for 2 days. After treatment, Sp1 levels were determined by Western blotting. (**C**) Green fluorescent protein (GFP)- or GFP-Sp1-overexpressing U87MG cells treated with BA for 2 days. Cell viability determined by MTT assay. Data presented as means ± standard deviations (t-Test: * *p* < 0.05, ** *p* < 0.01, and *** *p* < 0.005; ns: not significant). For more details on Western blot, please view Appendix A.

**Figure 4 cancers-12-00981-f004:**
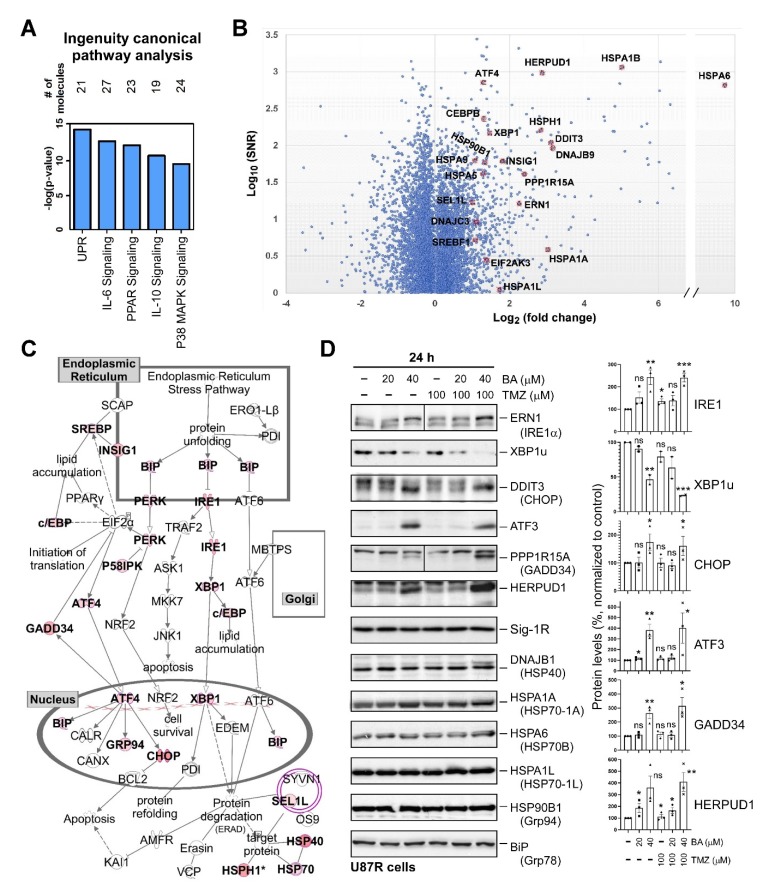
BA treatment alters expression of endoplasmic-reticulum (ER)-stress-related genes: (**A**) 1341 genes showing fold changes of greater than or equal to 2, and *p* values of less than or equal to 0.05 between BA and DMSO treatment used as input for canonical pathway analysis using IPA software. (**B**) Data for all genes plotted as log_2_ fold changes versus log_10_ of signal-to-noise ratio. Genes selected as significantly different and involved in unfolded-protein-response (UPR) signaling highlighted as red dots. (**C**) UPR pathway generated by IPA software. Colored molecules represent genes affected under BA treatment. Red molecules indicate upregulated genes. (**D**) U87MG cells collected 1 day after treatment with BA and/or TMZ at different concentrations. Cellular lysates were then analyzed by Western blotting with indicated antibodies. (Right panel) Protein levels of IRE1, XBP1u, CHOP, ATF3, GADD34, and HERPUD1 normalized to tubulin. Data presented as means ± standard deviations (t-Test: * *p* < 0.05, ** *p* < 0.01, and *** *p* < 0.005; ns: not significant). For more details of Western blot, please view Appendix A.

**Figure 5 cancers-12-00981-f005:**
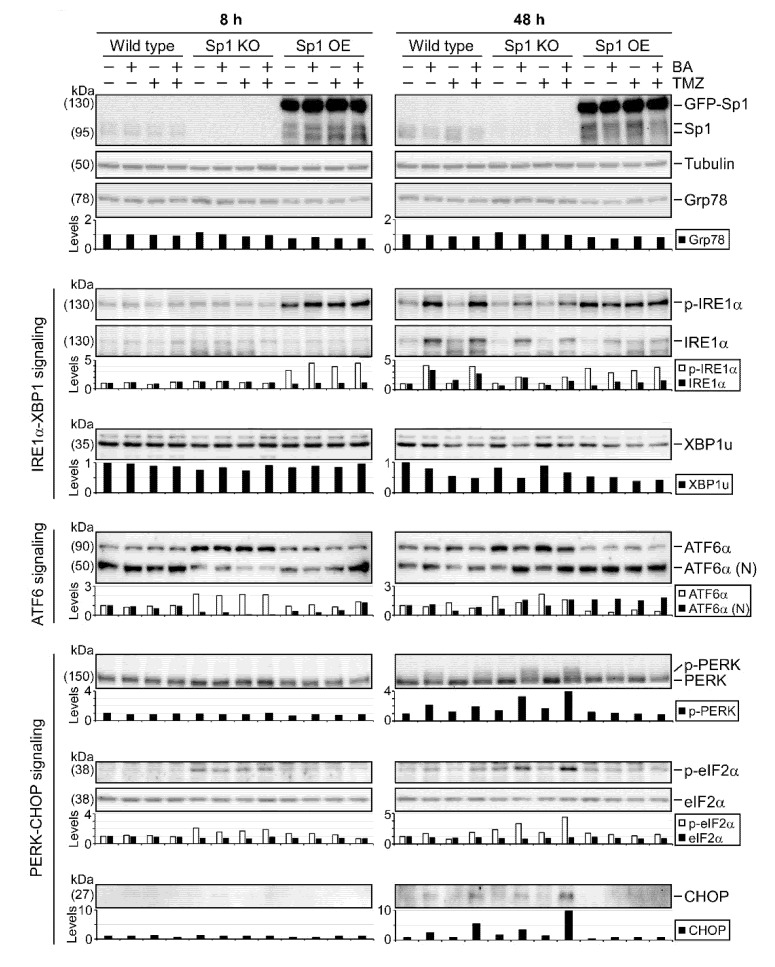
Roles of Sp1 in regulating UPR signaling. Wild-type, Sp1-knockout, and GFP-Sp1-overexpressing U87MG cells collected 8 h or 2 days after treatment with BA (20 μM) and/or TMZ (100 μM). Cellular lysates then analyzed by Western blotting with indicated antibodies. For more details of Western blot, please view Appendix A.

**Figure 6 cancers-12-00981-f006:**
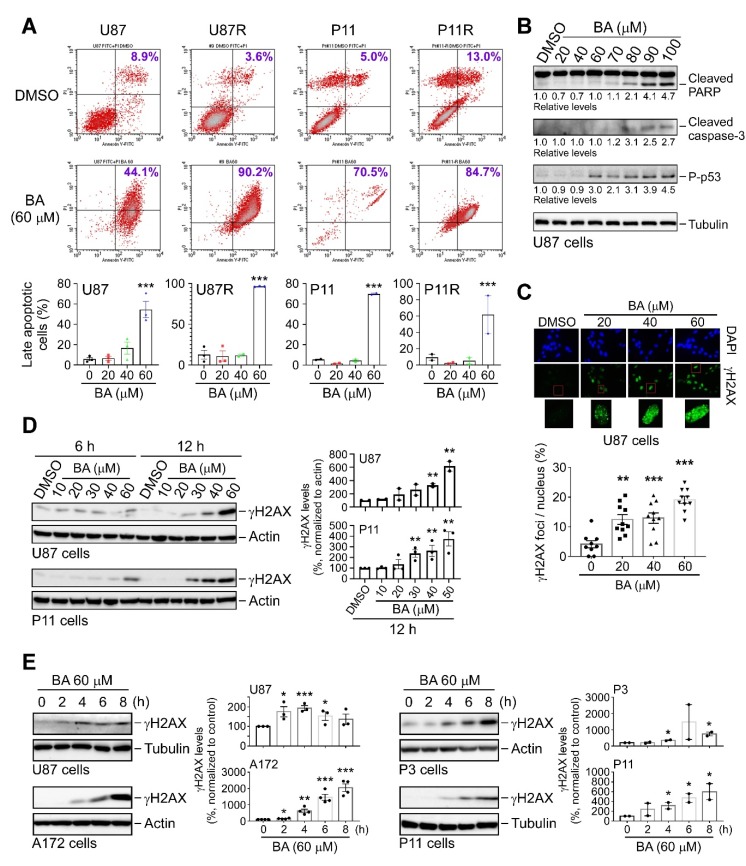
BA induces apoptosis and DNA damage in GBM cells. (**A**) Cells treated with DMSO or different concentrations of BA for 2 days. After treatment, cells were stained with both Annexin V-fluorescein isothiocyanate (FITC) and propidium iodide (PI), and apoptotic cell death was measured using flow cytometry. (Lower panels) Percentages of late apoptotic cells shown in histograms. (**B**) Cells collected 2 days after treatment with different concentrations of BA. Cellular lysates then analyzed by Western blotting with indicated antibodies. (**C**) Cells treated with different concentrations of BA for 24 h, and immunofluorescence images of γH2A.X staining in cells shown in green. Nuclei were stained with 4′,6-diamidino-2-phenylindole (blue). (**D**,**E**) Cells collected at various time points after initial treatment with different concentrations of BA. Cellular lysates then analyzed by Western blotting with indicated antibodies. Data presented as means ± standard deviations (t-Test: * *p* < 0.05, ** *p* < 0.01, and *** *p* < 0.005; ns: not significant). For more details of Western blot, please view Appendix A.

**Figure 7 cancers-12-00981-f007:**
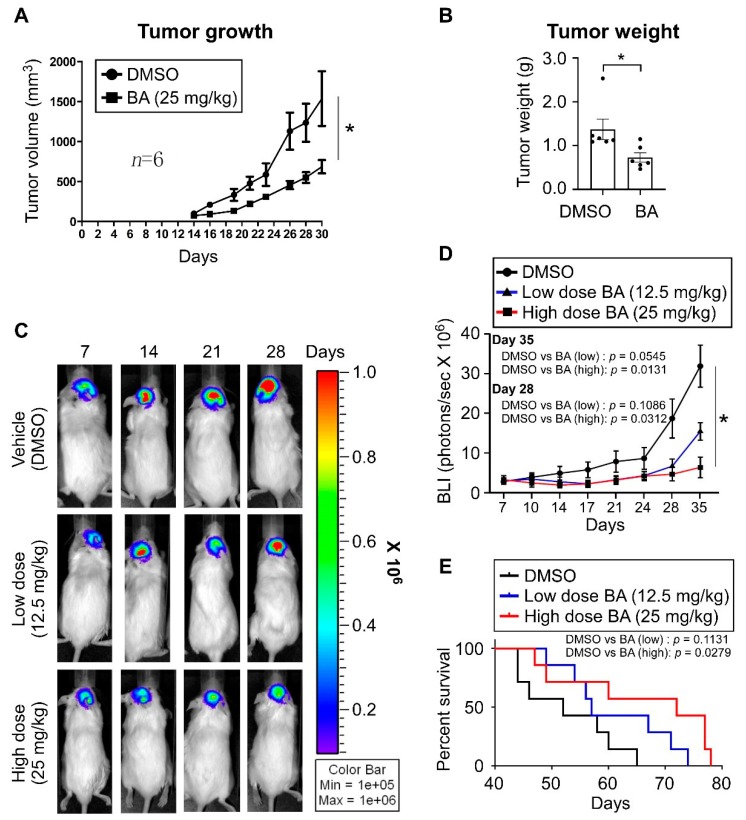
BA enhances therapeutic efficacy and long-term survival in GBM xenograft model mice. (**A**,**B**) TMZ-resistant U87MG cells implanted subcutaneously in NOD/SCID mice. On Day 4 after transplantation, mice were randomly grouped and treated with DMSO or BA for 4 weeks. (**A**) Tumor volume (length × width^2^ × 3.14/6) was measured three times a week. After treatment, mice were sacrificed. (**B**) Tumor weights. (**C**–**E**) Established orthotopic GBM xenograft animal model from luciferase + GFP dual-labeled U87MG cells (2.5 × 10^5^ cells/mouse). Mice were treated with DMSO or BA at different doses three times a week. (**C**,**D**) Tumor growth monitored using in vivo image system (IVIS) twice a week. (**E**) Survival plotted using Kaplan–Meier curves. (**A,B,D**) Data presented as means ± standard deviations (t-Test: * *p* < 0.05).

**Figure 8 cancers-12-00981-f008:**
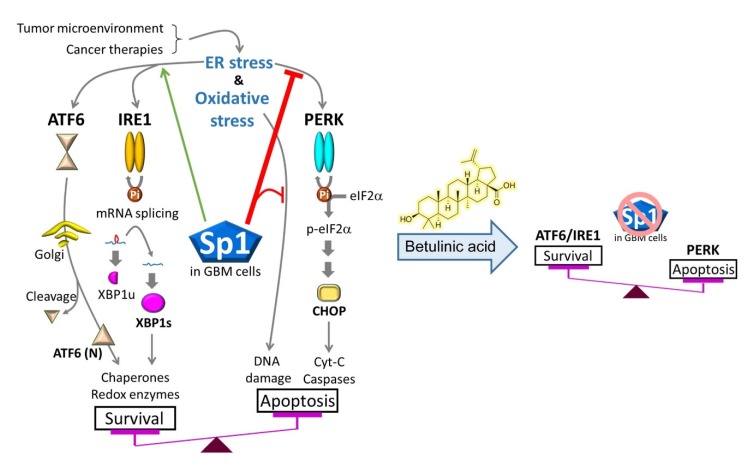
Sp1 protects malignant GBM cells against cancer therapies by altering ER/oxidative-stress responses, particularly inhibition of protein kinase RNA-like endoplasmic reticulum kinase (PERK)/eIF2α/ C/EBP-homologous protein (CHOP) activation. However, BA also reduces Sp1 expression, thereby promoting apoptotic cell death in GBM.

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
