# Peer review of "Betulinic Acid-Mediated Tuning of PERK/CHOP Signaling by Sp1 Inhibition as a Novel Therapeutic Strategy for Glioblastoma"

_cancers, 2020, doi:10.3390/cancers12040981_

Round 1
Reviewer 1 Report
The changes appear to be sufficient to address the reviewers concerns.
Reviewer 2 Report
The authors have addressed all of my concerns by doing appropriate experiments and update the manuscripts where necessary. The quality of the manuscript has been significantly improved. I have no further concern regarding this manuscript.
This manuscript is a resubmission of an earlier submission. The following is a list of the peer review reports and author responses from that submission.
Round 1
Reviewer 1 Report
Overall the manuscript is well written, and the conclusions are presented adequately in a concise manner. The main problem lies with the presentation of the data figures and some figures are missing some key elements that need to be addressed.
1). All immunoblots need densitometry with statistical analysis when making a claim in the results section that there is a difference between treatments.
2). All data figures need statistical analysis Such as in figure 7 D&E in order to make the claim that there was an actual difference in survival.
3). Immuno Fluorescence images also should have some kind of quantitative image analysis to show that on average ϒH2AX is actually increasing rather than just relying on representative images.
4 Figure 4 C is not very legible. The IPA software-generated image should be remade with a larger font, and with colors that do not clash with the font color.
Reviewer 2 Report
- In figure 1, it is unclear why two different types of graphs were used to show the same type of data. More importantly, the conclusion “BA selectively targets brain tumor cells” should not be made if the normal/cancer comparison was done only on U87 cells. Please repeat experiment in figure 1A with other types of cancer cells.
- In figure 2A, please determine statistically if BA has a synergistic or additive effect in combination with TMZ.
- What are the biological effects of Sp1 KO/OE? Do they match the effect of BA treatment?
- Experiments in figure 5 need to be repeated on at least another cell line.
- In figure 7E, please perform statistical analysis to compare survival of different treatment groups.
- The authors used normalized cell viability in many figures. The controls were normalized to 1, and there were no error bars included. Please calculate and add error bars to those samples.
- In column graphs, please plot all data points were available. https://www.graphpad.com/support/faq/graph-tip-how-can-i-make-a-barcolumn-graph-that-also-shows-the-individual-data-points/
- In all western blot figures, please crop the blot further away from the bands, especially on left and right side. Furthermore, please explain why some blots were divided in half (e.g. 4D, 6D, 6E).
- There are multiple typos throughout the manuscript, including in the figures. Please proofread the manuscript.
